# Natural and synthetic antimicrobials reduce adherence of enteroaggregative and enterohemorrhagic *Escherichia coli* to epithelial cells

Yaraymi Ortiz[1], Alam García-Heredia[2], Angel Merino-Mascorro[1], Santos García[1], Luisa Solís-Soto[1], Norma Heredia[1]*

1 Facultad de Ciencias Biológicas, Departamento de Microbiología e Inmunología, Ciudad Universitaria, Universidad Autónoma de Nuevo León, San Nicolás de los Garza, Nuevo León, México, 2 Molecular and Cellular Biology Graduate Program, University of Massachusetts, Amherst, MA, United States of America

* norma@microbiosymas.com

**Data Availability Statement:** All relevant data are within the paper and its Supporting Information files.

## Abstract

Adherence of bacteria to the human intestinal mucosa can facilitate their internalization and the development of pathological processes. *Escherichia coli* O104:H4 is considered a hybrid strain (enteroaggregative hemorrhagic *E. coli* [EAHEC]), sharing virulence factors found in enterohemorrhagic (EHEC), and enteroaggregative (EAEC) *E. coli* pathotypes. The objective of this study was to analyze the effects of natural and synthetic antimicrobials (carvacrol, oregano extract, brazilin, palo de Brasil extract, and rifaximin) on the adherence of EHEC O157:H7, EAEC 042, and EAHEC O104:H4 to HEp-2 cells and to assess the expression of various genes involved in this process. Two concentrations of each antimicrobial that did not affect (p≤0.05) bacterial viability or damage the bacterial membrane integrity were used. Assays were conducted to determine whether the antimicrobials alter adhesion by affecting bacteria and/or alter adhesion by affecting the HEp-2 cells, whether the antimicrobials could detach bacteria previously adhered to HEp-2 cells, and whether the antimicrobials could modify the adherence ability exhibited by the bacteria for several cycles of adhesion assays. Giemsa stain and qPCR were used to assess the adhesion pattern and gene expression, respectively. The results showed that the antimicrobials affected the adherence abilities of the bacteria, with carvacrol, oregano extract, and rifaximin reducing up to 65% (p≤0.05) of *E. coli* adhered to HEp-2 cells. Carvacrol (10 mg/ml) was the most active compound against EHAEC O104:H4, even altering its aggregative adhesion pattern. There were changes in the expression of adhesion-related genes (*aggR*, *pic*, *aap*, *aggA*, and *eae*) in the bacteria and oxidative stress-related genes (*SOD1*, *SOD2*, *CAT*, and *GPx*) in the HEp-2 cells. In general, we demonstrated that carvacrol, oregano extract, and rifaximin at sub-minimal bactericidal concentrations interfere with target sites in *E. coli*, reducing the adhesion efficiency.

**Funding:** This support was granted to NH by the Consejo Nacional de Ciencia y Tecnología de Mexico (CONACYT, www.conacyt.gob.mx) [CB-2016-01, Proy # 285151], and by Universidad Autónoma de Nuevo León [Programa de Apoyo a la Investigación Científica y Tecnológica, PAICYT, http://investigacion.uanl.mx/paicyt-provericyt/]. Yaraymi Ortiz had scholarship granted from CONACYT, and Alam García-Heredia had scholarship granted from UANL.

**Competing interests:** The authors have declared that no competing interests exist.

## Introduction

Although *Escherichia coli* is part of the commensal microbiota in warm-blooded animals, several strains can cause health problems. These are grouped into six pathotypes: enteropathogenic (EPEC), enterotoxigenic (ETEC), enterohemorrhagic (EHEC), enteroinvasive (EIEC), diffusely adherent (DAEC), and enteroaggregative (EAEC), [1].

Various pathogenic *E. coli* strains in the human intestine use an infection strategy that involves adhesion to target cells, colonization of the mucosal site, evasion of host defenses, multiplication, and damage to the host. Colonization of the intestinal mucosa is the most conserved ability among diarrheal strains of *E. coli* [2]. Among the pathogenic traits, the initial adhesion is critical to establish a successful infection. Strains of the EAEC pathotype (named due to their aggregative pattern of adherence to HEp-2 cells) have been implicated in outbreaks of watery diarrhea (particularly in travelers) in various parts of the world [3]. When these aggregative strains adhere to intestinal cells in humans, they form bacterial aggregates resembling stacked bricks. This pattern is due to virulence factors encoded on the aggregative adherence plasmid (pAA), including *aggA* (which encodes a major component of the aggregative adherence fimbriae [AAF]) and *aap* (which encodes the surface protein dispersin, which partially counteracts aggregation) [4].

Strains of the EHEC pathotype cause diseases from self-limiting diarrhea to more serious pathologies such as hemolytic uremic syndrome [5, 6]. These strains can produce the proteins Shiga toxin 1 (Stx1) and/or Shiga toxin 2 (Stx2). They also have additional virulence factors, such as intimin, translocated intimin receptor, and a type three secretion system (SST3). Genes that codify these virulence factors are located in the locus of enterocyte effacement (LEE) pathogenicity island, and they underlie the localized adherence phenotype [7–9].

Important outbreaks of hemorrhagic diarrhea caused by the O104:H4 strain, which has an aggregative pattern of adherence, occurred in 2011 in several countries [10]. This strain produces Stx2a, which is generally associated with EHEC strains [11]. Due to this, the O104:H4 strain was placed in a new lineage known as enteroaggregative hemorrhagic *E. coli* (EAHEC), which affects humans [12]. Aggravated sequelae were observed, suggesting that, in this hybrid strain, Stx2a crosses the intestinal mucosa more efficiently and/or in greater quantities, perhaps due to simultaneous colonization of multiple intestinal segments or due to improved intestinal colonization that facilitates the systemic absorption of Stx2 [5, 13]. However, the exact reason remains unclear [14].

EAEC binds to the intestinal surface and induces increased mucus secretion. The excess of mucus traps the bacteria, promoting aggregate formation, virulence, long-term persistence, and the action of secreted enterotoxins and cytotoxins [15].

EAHEC O104:H4 has high rates of antibiotic resistance [16]. Additionally, antibiotics can lead to increased toxicity during the prodromal phase of diarrhea via the Jarisch–Herxheimer reaction, involving massive Stx release due to bacterial death [17]. To make this worse, is evidenced that sub-lethal concentrations of aminoglycoside antibiotics induce biofilm formation in clinical isolates of *Pseudomonas aeruginosa* and *E. coli* further suggesting that stimulate bacterial adhesion to the host cells [18, 19].

The use of natural antimicrobials as food preservatives or as therapeutics has been regarded as a potential alternative to reduce these negative effects. Previously, our group demonstrated that oregano (*Lippia graveolens*) extract and palo de Brasil (*Haematoxylum brasiletto*) extract (Hb extract) and their major compounds, carvacrol and brazilin, respectively, affected the growth and swarming motility of *E. coli* pathotypes (including O104:H4) and upregulated virulence genes [20]. We reasoned that these antimicrobials may ameliorate the cell adhesion ability of these pathogenic strains. Thus, the aim of this study was to analyze the adhesion

process and virulence gene expression in EAEC, EHEC, and the hypervirulent serotype O104:
H4 by using subinhibitory concentrations of natural antimicrobials and the antibiotic rifaxi-
min, which was the recommended therapy used during the major O104:H4 outbreak [21]. We
explored various scenarios in which the antimicrobials may affect bacterial adhesion during
infection: the effects of the antimicrobials on the bacteria, on the HEp-2 cells, and on already
established adhesion between the bacteria and HEp-2 cells. We found that the antimicrobials
inhibit adhesion by directly acting on *E. coli*. This effect remained present to some degree even
after three cycles of adherence assays. Our findings suggest that the antimicrobials could
decrease the virulence of pathogenic *E. coli*.

## Materials and methods

### Bacteria and culture conditions

EAHEC serotype O104:H4 was provided by Penn State University culture collection. EAEC
serotype 042 was kindly provided by Dr. Fernando Navarro (CINVESTAV-Mexico). EHEC
serotype O157:H7 (American Type Culture Collection [ATCC] 43894) was kindly provided by
Dr. Lynn McLasborough (Food Science Dept, University of Massachusetts, Amherst, MA,
USA). These three strains are hereafter referred to as EAHEC, EAEC, and EHEC, respectively.
The strains were maintained at -80˚C in Brain Heart Infusion (BHI) broth (Bioxon, Becton-
Dickinson, Franklin Lakes, New Jersey, USA) with 20% glycerol (Sigma-Aldrich, Mexico).
Each active culture was prepared by transferring an aliquot into a tube containing 16 ml fresh
BHI agar and incubating for 48 h at 37˚C. Cultures were stored at 4˚C for no more than 8
weeks. For the assays, an aliquot was transferred into a tube containing 5 ml Mueller Hinton
(MH) broth (Difco, Becton-Dickinson, Franklin Lakes, New Jersey, USA) or Eagle's Minimum
Essential Medium (EMEM; ATCC® 30–2003™; ATCC, Manassas, VA, USA) and incubated
overnight at 37˚C.

### Antimicrobials

To create the oregano and Hb extracts, leaves of oregano (*Lippia graveolens* Kunth) and bark
of palo de Brasil (*Haematoxylum brasiletto* Karsten) were obtained from local markets. They
were identified by the Department of Botany, Universidad Autónoma de Nuevo León (San
Nicolás, Nuevo León, México). Ethanolic extracts were prepared by mixing 100 g of ground
material with 500 ml of 96% ethanol (CTR Scientific, Monterrey, Mexico). The extracts were
macerated at room temperature for 24 h and then filtered through Whatman no. 1 paper. The
liquid fraction was placed on glass plates for 48–72 h until complete ethanol evaporation [20].
The dried extracts were resuspended in 96% ethanol and maintained at 4˚C in amber flasks for
no more than 12 weeks. An aliquot was used to determine the dry weight and working solu-
tions were prepared at the beginning of each experiment. The same batch of oregano and palo
de Brasil was always used. These were preserved as dried material, and under controlled condi-
tions of light (darkness), humidity and temperature. When a fresh extract was prepared, the
MBC (minimum bactericidal concentration) was determined; no variation on MBC of the
extracts were found.

Carvacrol (W224511; Sigma Aldrich, Mexico) and brazilin (154862; MP Biomedicals, Ill-
kirch-Graffenstaden, France) were dissolved in water, with 0.05% Tween 20 in the case of carva-
crol. Rifaximin ($C_{43}H_{51}N_3O_{11}$; R9904; Sigma-Aldrich, San Luis Missouri, USA) was dissolved
in 96% ethanol. They were maintained as stock solutions in amber flasks at 4˚C for no more
than 2 weeks, and working solutions were prepared at the beginning of each experiment.

The MBC of each antimicrobial was previously determined using a standard broth microdi-
lution method [20] and confirmed in this study.

## Cell line

The human cervical carcinoma cell line HEp-2 (ATCC® CCL-23™), derived from epithelial cells, was purchased from ATCC (Manassas, Virginia, USA) and maintained in liquid nitrogen. An aliquot was transferred to a 25-cm$^2$ plastic tissue culture flask (Corning Costar, Corning, Nueva York, USA) containing 3 ml EMEM (ATCC® 30–2003™; ATCC) supplemented (EMEMsup) with 1% nonessential amino acids, 1% penicillin–streptomycin (GIBCO, Thermo Scientific, Waltham, Massachusetts, USA), and 10% fetal bovine serum (Corning). The HEp-2 cells were incubated at 37˚C in 5% $CO_2$ (Steri-Cycle $CO_2$ Incubator; Thermo Scientific) and used between passages 7 and 30.

## Effects of antimicrobials on bacterial and HEp-2 cell viability

**a) Effects on bacteria.** The effects on bacterial viability of antimicrobial concentrations lower than the MBC (sub-MBC; S2 Table) was determined. Bacteria were inoculated (2 ml, $2\times10^7$ CFU/ml) in tubes containing 18 ml MH broth or EMEM plus various antimicrobial concentrations. The cultures were incubated at 37˚C with shaking (150 rpm; MaxQ Mini 4450; Barnstead/Lab-Line, Melrose Park, Illinois) for 12 h. Aliquots were taken every 2 h, and bacteria were enumerated using the plate count method in MH agar. The controls were bacteria grown in medium (MH broth or EMEM) with water instead of antimicrobials, and medium with the antimicrobial but without the bacteria. The pH of the medium with each antimicrobial was also measured (Hanna HI9023 pH meter; Hanna Instruments, Woonsocket, Rhode Island, USA).

Damage to the bacteria was determined by assessing the bacterial membrane integrity. For this, bacteria were grown overnight, adjusted to $5\times10^8$ CFU/ml, and exposed to antimicrobials at sub-MBC for 1 and 4 h (durations of the adhesion assays) at 37˚C with shaking (150 rpm). Following the manufacturer's recommendations with modifications [22], a LIVE/DEAD kit (Invitrogen, Thermo Scientific, Waltham, Massachusetts, USA) was then used with fluorescence flow cytometry (Attune® Acoustic Focusing Cytometer; Applied Biosystems, Thermo Scientific, Waltham, Massachusetts, USA) at excitation/emission maxima of 535/617 nm employing a Blue Laser 3 (BL3) detector. All experiments were evaluated based on 30,000 events.

**b) Effects on HEp-2 cells.** The effects on HEp-2 cell viability of antimicrobials at sub-MBC was determined using 3-(4,5-dimethylthiazol-2-yl)-2,5-diphenyltetrazolium bromide (MTT) assays as described by Mosmann [23] with modifications. Briefly, the cells were seeded ($1\times10^4$ cells) in 96-well tissue culture plates containing 200 μl EMEMsup and incubated at 37˚C in 5% $CO_2$ for 48 h (allowing the cells to reach confluence). The medium was removed and 200 μl of fresh medium containing antimicrobials at sub-MBC were added, and the plates were incubated again for 4, 12, and 24 h at 37˚C in 5% $CO_2$. At each timepoint, the cell morphology was observed using an inverted microscope (40X Olympus CK40; Shinjuku, Tokyo, Japan). For the MTT assay, the medium was removed by aspiration. Next, 90 μl Roswell Park Memorial Institute (RPMI)-1640 medium without phenol red (Sigma-Aldrich) but with 10% fetal bovine serum and 10 μl of 5 mg/ml MTT (Sigma-Aldrich) was added to each well. After 3 h of incubation, the MTT solution was removed and 100 μl isopropanol (Sigma-Aldrich) with 0.04 N HCl was added for formazan solubilization. Absorbance at 570 nm was assessed using a microplate spectrophotometer (Epoch 2; BioTek Instruments, Winooski, Vermont, USA).

## Effects of antimicrobials on *E. coli* adhesion to HEp-2 cells

The effects of antimicrobials on the ability of bacteria to adhere to HEp-2 cells were analyzed as described by Cravioto [24] with modifications. Monolayers of confluent HEp-2 cells (in

EMEMsup in 25-cm$^2$ plastic tissue culture flasks) were dispersed with 0.5 ml trypsin solution (0.25% w/v; Sigma-Aldrich), counted in a Neubauer chamber (Vela Quin, Iztapalapa, Ciudad de México, Mexico), and diluted with EMEMsup. For the adherence assays, $2\times10^5$ HEp-2 cells were placed into 24-well plates and incubated at 37˚C in 5% $CO_2$ until they reached confluence. The culture medium was changed when its color changed (48–72 h), due to acidification of pH by cellular growth. The following adherence assays were performed to elucidate the antimicrobial targets (S1 Fig):

**a) Assay of the effects of antimicrobials on bacteria and, to a lesser extent, on HEp-2 cells (S1A Fig).**   A 50-µl aliquot of fresh *E. coli* culture was inoculated in 5 ml EMEM and incubated overnight at 37˚C. Cultures were adjusted to $5\times10^7$ CFU/ml ($A_{600} = 0.065$) with EMEM and the antimicrobial at sub-MBC was added. After 1 h at 37˚C with shaking (150 rpm), 200 µl of whole culture (containing bacteria, the medium, and the antimicrobial) was inoculated into wells with confluent HEp-2 cells (100:1 bacteria:HEp-2 cells). After incubation at 37˚C in 5% $CO_2$ for 3 h, the plates were washed with phosphate-buffered saline (PBS; 0.01 M; pH 7.4) to remove non-adhered bacteria. The number of adhered bacteria was determined by adding 200 µl of 0.5% sodium deoxycholate (Sigma-Aldrich) to detach the bacteria from the cells. Detached bacteria were 10-fold serially diluted, plated on MH agar, and counted after incubation at 37˚C for 48 h. The adherence pattern was determined by Giemsa staining. Samples of adhered and non-adhered bacteria were collected for gene expression assays.

**b) Assay of the effects of antimicrobials on bacteria only (S1B Fig).**   To assess whether the antimicrobials in the bacterial culture played a role in any effects on adhesion via affecting the Hep-2 cells, we conducted assays that involved removing the antimicrobials prior to inoculating the Hep-2 cells with the antimicrobial-treated bacteria. An aliquot of fresh *E. coli* culture was inoculated in 5 ml MH broth or EMEM and incubated overnight at 37˚C. Cultures were adjusted to $5\times10^7$ CFU/ml on MH broth or EMEM and the antimicrobial at sub-MBC was added. After incubation at 37˚C with shaking (150 rpm) for 1 h, the cultures were centrifuged at 2,700*g* for 10 min at room temperature to remove the antimicrobial. The bacteria were washed twice with PBS and adjusted to $2\times10^8$ CFU/ml ($A_{600} = 0.085$) in EMEM without supplements. Next, 200 µl was added to 24-well plates containing confluent HEp-2 cells (100:1 bacteria:HEp-2 cells). After incubation at 37˚C in 5% $CO_2$ for 3 h, the plates were washed, and non-adhered bacteria were removed. The number of adhered bacteria was determined, as above.

**c) Assay of the effects of antimicrobials on HEp-2 cells only (S1C Fig).**   To assess whether incubation of HEp-2 cells with each antimicrobial would protect against bacterial adhesion, confluent HEp-2 cells in 24-well plates were washed twice with PBS and 200 µl EMEM containing the antimicrobial at sub-MBC was added. After 1 h of incubation at 37˚C in 5% $CO_2$, the medium was removed and 200 µl bacterial culture ($2\times10^8$ CFU/ml) grown in EMEM was added (100:1 bacteria:HEp-2 cells) and incubated at 37˚C in 5% $CO_2$ for 3 h. Following washing to remove non-adhered bacteria, the number of adhered bacteria was determined as above.

**d) Assay of the effects of antimicrobials on already established bacteria–HEp-2 cell adhesion (S1D Fig).**   Plates containing confluent HEp-2 cells were inoculated with bacteria in EMEM (100:1; bacteria:HEp-2 cells). After incubation at 37˚C in 5% $CO_2$ for 1 h, 200 µl antimicrobial at sub-MBC (resuspended in EMEM) was added and incubated at 37˚C in 5% $CO_2$ for 3 h. Following washing to remove non-adhered bacteria, the number of adhered bacteria was determined as above.

**e) Assay of the duration of adherence ability exhibited by non-adhered antimicrobial-treated bacteria (S1E Fig).**   A question arose regarding whether bacteria that were exposed to each antimicrobial, and subsequently did not adhere to HEp-2 cells, would continue not to be

able to adhere and, if so, would this last across several generations? Thus, non-adhered bacteria (previously exposed to the antimicrobial at sub-MBC, followed by the adhesion assay) from experiment *a*) were washed twice with PBS and adjusted to $2\times10^8$ CFU/ml with EMEM. A 200-μl aliquot was added to confluent HEp-2 cells (100:1 bacteria:HEp-2 cells). After incubation at 37˚C in 5% $CO_2$ for 3 h, the supernatant containing non-adhered bacteria was collected and used to repeat the same experiment another two times. In each cycle, the number of adhered bacteria was determined by the plate count method in MH agar.

### Gene expression analysis

The effects of rifaximin, carvacrol, and oregano extract (which were found to reduce bacterial adhesion) on the expression of *aggR* (putative transcriptional activator), *aap* (dispersin), *aggA* (a major component of the AAF), and *pic* (secreted protease) by EAHEC and EAEC, *eae* (intimin) by EHEC, and *stx2a* by EAHEC and EHEC (S1 Table), using adhered and non-adhered bacteria (from the adhesion assays), were analyzed. Additionally, the expression of genes encoding enzymes related to oxidative stress in HEp-2 cells after antimicrobial treatment with/without EAHEC/EAEC was evaluated (S1 Table).

In all cases, RNA extraction was performed using TRIzol reagent (Bioline, Meridian Bioscience, Cincinnati, Ohio, USA). RNA integrity was determined using a UV-Vis spectrophotometer (NanoDrop 2000, Thermo Scientific) at 260 nm. cDNA synthesis was performed using an iScript™ Synthesis Kit (Bio-Rad, Hercules, California, USA) and gene expression was quantified by real-time PCR (PikoReal Real-time PCR System, Thermo Scientific) using Q SYBR Green Supermix (Thermo Scientific). The cycling conditions were 95˚C/3 min followed by 40 cycles at 95˚C/15 s and 60˚C/30 s.

The results were normalized using the housekeeping gene 16S rRNA for the bacteria and 18S rRNA for the HEp-2 cells, which remained their expression levels stable in the presence of the antimicrobials for both bacteria (Cq: 21.02±3.34) and HEp-2 cells (18.74±2.06) with 4 and 3 h of exposure, respectively. The relative fold change was calculate using the ΔΔ cycle threshold method [25]. Thus, the results were expressed as relative fold change in expression of the antimicrobial-treated group compared to the relevant control group (HEp-2 cells and bacteria untreated).

The mean ΔCq values (mean of the triplicate for each sample) were compared in each case. The relative quantification of mRNA copies was compared with untreated control samples by analysis of variance (ANOVA) and Dunnett's post-hoc test.

### Statistical analyses

The statistical analyses were performed using IBM SPSS Statistics version 20.0 (IBM Corp, Armonk, NY, USA), and Origin version 9.0 (Northampton, Massachusetts, USA) was used for graphics. Using ANOVA was compared the mean results for different antimicrobial concentrations by Duncan's post-hoc tests, and the mean results for the antimicrobials and control also by Dunnett's post-hoc tests. For each assay, three separate experiments were conducted with at least three replicates each. $p \leq 0.05$ was considered significant.

## Results

### Antimicrobials affect bacterial viability

To study the effects of the antimicrobials on *E. coli* adhesion, we first verified the MBC of each antimicrobial against the *E. coli* strains. We found that EAEC was the most sensitive strain to most of the antimicrobials (S1 Table). Rifaximin exerted the highest inhibitory effect (MBC:

0.02–0.03 mg/ml), followed by carvacrol (MBC: 0.05–0.06 mg/ml), oregano extract (MBC: 0.70–0.85 mg/ml), brazilin (MBC: 2.65–3.2 mg/ml), and Hb extract (MBC: 3.8–4.3 mg/ml). The oregano and Hb extracts had higher MBCs ($p \leq 0.05$) than their major compounds (S2 Table).

We wondered whether the pH of the MH broth or EMEM changed after adding the antimicrobials, which may account for the bactericidal effects. Although several significant differences were observed (S3 Table), the pH values of the media were 7–8, with exception of pH 8.1 ±0.1 for 0.005 mg/ml rifaximin. These pH values should not affect the viability of either the bacteria or the HEp-2 cells.

Having established the MBCs, we selected sub-MBCs that would allow us to determine the effects of the antimicrobials on live *E. coli* adhering/adhered to HEp-2 cells. We initially selected two sub-MBC for each antimicrobial (S2 Table) and found that these concentrations did not affect bacterial growth during the exponential or stationary phases in either MH broth or EMEM, with the exception of 0.01 mg/ml rifaximin. Therefore, lower concentrations (0.005 and 0.002 mg/ml) were analyzed, though 0.005 mg/ml rifaximin still reduced the viability ($p \leq 0.05$) of EAEC during the exponential phase. Bacterial membrane integrity (evaluated by flow cytometry) supported these viability results (S4 Table). As a result, we selected the following sub-MBCs: 0.005 and 0.002 mg/ml for rifaximin; 0.025 and 0.010 mg/ml for carvacrol; 0.40 and 0.20 mg/ml for oregano extract; 1.5 and 1.0 mg/ml for brazilin; and 3.0 and 1.5 mg/ml for Hb extract.

The sub-MBCs selected for most antimicrobials did not affect ($p \leq 0.05$) the HEp-2 cell viability (with <10% reductions in viability in most cases) [26] after 3 h (which was the incubation duration in the adhesion assays) (S5 Table). However, as carvacrol at 0.025 mg/ml reduced ($p \leq 0.05$) the HEp-2 cell viability by 70%, it was not used in the subsequent assays (only 0.010 mg/ml carvacrol was used). An important finding was that rifaximin (at both sub-MBCs) and brazilin (at both sub-MBCs) actually increased HEp-2 cell viability at 4 h of exposure. However, at 12 or 24 h, the viability was similar to in the control (S5 Table).

## Preincubation with antimicrobials alters EHEC, EAEC, and EAHEC adhesion to HEp-2 cells

We first wondered whether preincubation of the bacteria with antimicrobials could reduce their adhesion ability. To explore this, we grew each *E. coli* strain individually and then added the antimicrobials at sub-MBC. After 1 h of incubation, the whole mixture (containing bacteria, the medium, and the antimicrobial) was added to HEp-2 cells. We found that rifaximin, carvacrol, and oregano extract decreased bacterial adhesion to HEp-2 cells by 26.1% ($p \leq 0.05$) to 64.5% ($p \leq 0.05$) (Fig 1). EAEC was the most sensitive strain to rifaximin and oregano extract (0.40 mg/ml), whereas EHEC was the most resistant to all treatments. In contrast to rifaximin, carvacrol, and oregano extract, Hb extract and brazilin actually increased the adherence (causing hyper-adherence) of the two aggregative strains (EAEC 042 and EAHEC O104:H4) to HEp-2 cells, with more adherence than in the control group ($\leq 112.3\%$; $p \leq 0.05$). EHEC showed hyper-adherence ($\leq 103.5\%$) by presence of Hb extract.

Observation of the aggregative adherence pattern of the aggregative strains (EAEC 042 and EAHEC O104:H4) treated with antimicrobials at sub-MBC indicated different levels of disturbances in the characteristic stacked brick structure; the pattern was completely altered for EAHEC (Fig 2, S6 Table),[27] after treatment with carvacrol (0.010 mg/ml), where bacteria were located individually, having lost their ability to stack on top of each other (Fig 2C). This anti-aggregative phenotype in EAHEC adhesion was not observed with the other antimicrobials or strains analyzed. The change in bacteria–bacteria adhesion could be due to the observed

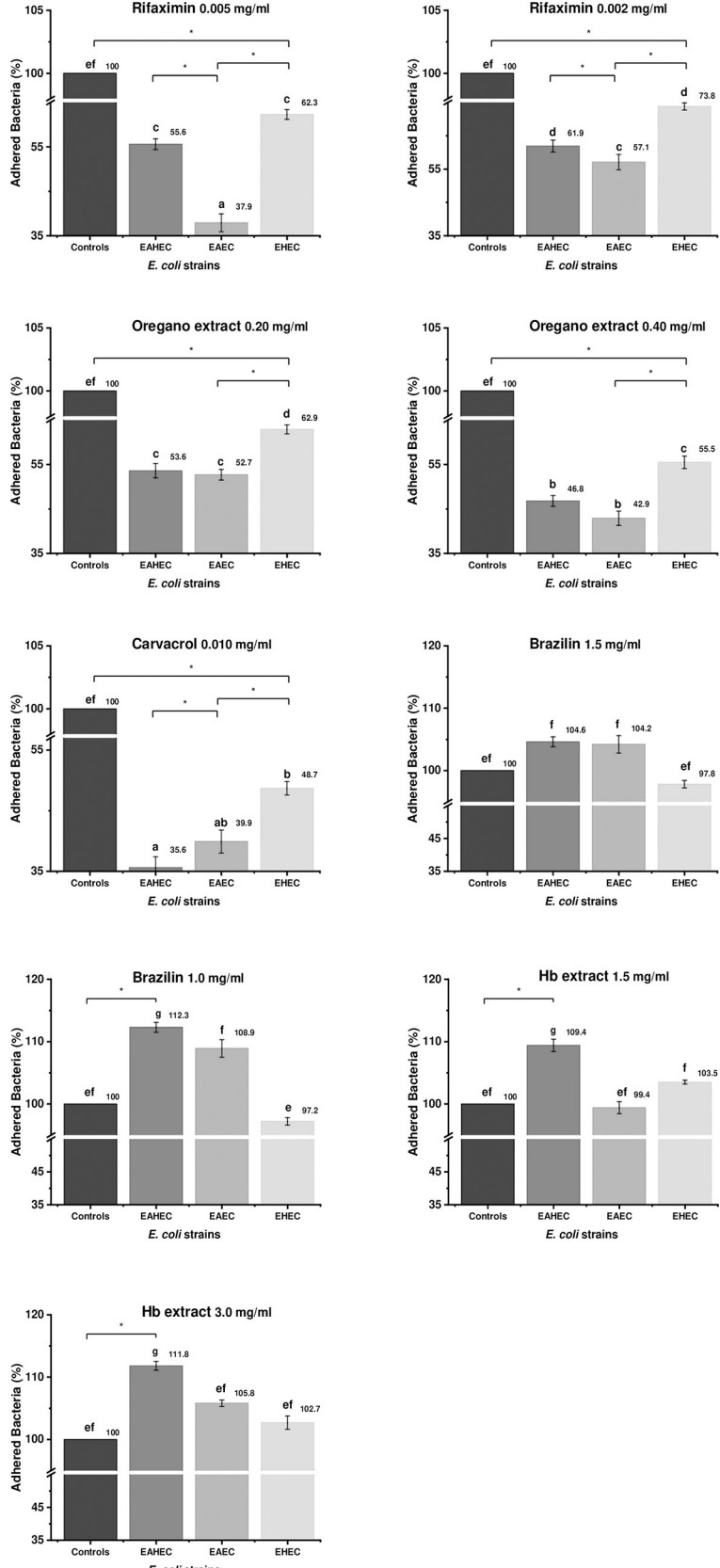

**Fig 1. Effects of antimicrobials on percentage of *E. coli* adhered to HEp-2 cells.** Bacteria were preincubated with antimicrobials for 1 h and the medium (with bacteria and the antimicrobial) was used to infect HEp-2 monolayers for 3 h. Different letters indicate significant differences from the control. Each strain was analyzed individually.

changes in gene expression in EAHEC (downregulation of *aggR*, *pic*, and *aap*, and upregulation of *aggA*), inducing the EAHEC to maintain (although at a reduced level) their ability to adhere to the target cell via the AAF (reflected by *aggA* upregulation). Therefore, the inconsistent expression of *aggA* and *aap* alters the bacteria–bacteria adhesion ability.

To explore whether the reduced adhesion to HEp-2 cells induced by antimicrobials was due to a change in gene expression, the expression of adhesion-related genes in the two aggregative strains (EAEC 042 and EAHEC O104:H4) was analyzed. Gene expression analysis of the bacteria adhered to HEp-2 cells showed that the two aggregative strains were both affected, with 0.002 mg/ml rifaximin causing upregulation of at least one of the adhesion-related genes (Table 1), more so in EAHEC. Additionally, in EAEC (for which bacteria–bacteria aggregation was not very affected), carvacrol highly upregulated both *aggR* and *pic* (50- and 31-fold, respectively).

The *stx2a* gene, encoded by the Stx2a phage, was downregulated by all antimicrobial treatments, except 0.40 mg/ml oregano extract treatment of adhered EHEC and EAHEC, which exhibited upregulation by 2.6-fold (p≤0.05) and 1.3-fold (p≤0.05), respectively. Additionally,

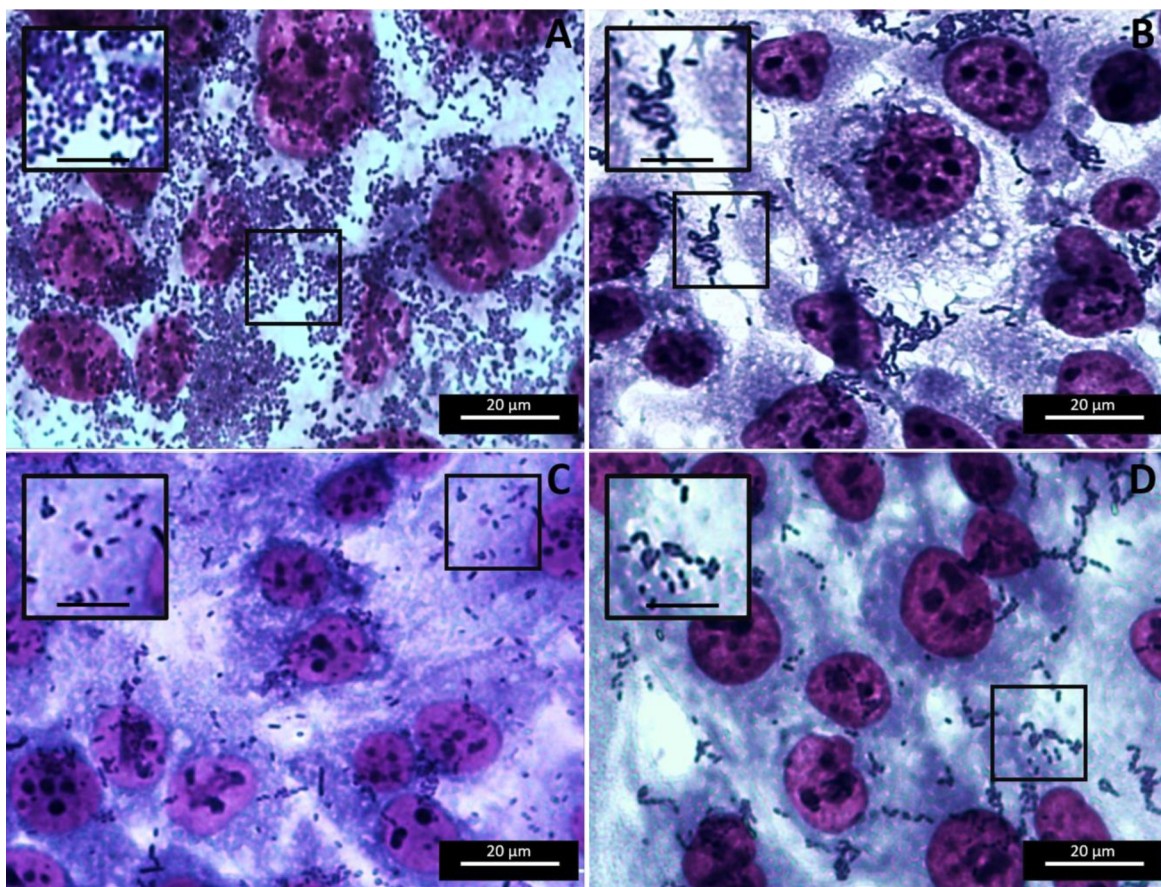

**Fig 2. *E. coli* O104:H4 adhered to HEp-2 cells.** *E. coli* grown for 1 h in EMEM with an antimicrobial (rifaximin, carvacrol, or oregano extract) at sub-MBC were used to infect HEp-2 cells for 3 h and followed by washing and staining with Giemsa. (A) Control, (B) rifaximin (0.005 mg/ml), (C) carvacrol (0.010 mg/ml), and (D) oregano extract (0.040 mg/ml).

**Table 1. Effects of antimicrobials on the fold changes in the expression of adhesion-related genes in *E. coli* strains.**

| Gene | *E. coli* serotype | Rifaximin (mg/ml) 0.005 | Rifaximin (mg/ml) 0.002 | Carvacrol (mg/ml) 0.010 | Oregano extract (mg/ml) 0.40 | Oregano extract (mg/ml) 0.20 |
|---|---|---|---|---|---|---|
| *aggR* | EAEC 042 | 0.0±0.0$^{a*}$ (2.5±0.3$^{d*}$) | 0.3±0.0$^{b*}$ (0.5±0.4$^{b*}$) | 50±0.0$^{g*}$ (4.6±0.3$^{e*}$) | 2.2±0.0$^{d*}$ (2.7±0.0$^{d*}$) | 2.6±0.1$^{e*}$ (10±0.1$^{f*}$) |
| | EAHEC O104:H4 | 7.7±0.7$^{e*}$ (75±0.8$^{g*}$) | 6.4±0.2$^{e*}$ (1.1±0.1$^{c}$) | 0.3±0.1$^{b*}$ (1.7±0.4$^{d*}$) | 0.6±0.5$^{b*}$ (7.1±0.3e$^{*}$) | 1.6±0.7$^{d*}$ (5.3±0.3$^{e*}$) |
| *pic* | EAEC 042 | 0.6±0.1$^{b}$ (0.4±0.0$^{b*}$) | 38±0.0$^{f*}$ (0.1±0.2$^{a*}$) | 31±0.1$^{f*}$ (16±0.0$^{f*}$) | 0.1±0.0$^{a*}$ (2.1±0.0$^{d*}$) | 0.1±0.3$^{a*}$ (2.8±0.0$^{e*}$) |
| | EAHEC O104:H4 | 5.0±0.0$^{e*}$ (174±0.5$^{g*}$) | 29±0.2$^{f*}$ (1.3±0.1$^{c}$) | 0.0±0.0$^{a*}$ (1.2±0.5$^{c}$) | 0.1±0.0$^{a*}$ (0.4±0.1$^{b*}$) | 0.1±0.0$^{a*}$ (0.4±0.1$^{b*}$) |
| *aggA* | EAEC 042 | 0.0±0.0$^{a*}$ (0.1±0.0$^{a*}$) | 0.0±0.0$^{a*}$ (0.0±0.0$^{a*}$) | 0.0±0.0$^{a*}$ (0.0±0.0$^{a*}$) | 0.0±0.0$^{a*}$ (0.0±0.0$^{a*}$) | 0.6±0.0$^{b*}$ (0.2±0.0$^{a*}$) |
| | EAHEC O104:H4 | 2.3±0.0$^{d*}$ (1.2±0.1$^{c}$) | 2.7±0.0$^{e*}$ (0.02±0.0$^{a*}$) | 3.0±0.1$^{e*}$ (0.7±0.0$^{b*}$) | 3.4±0.1$^{e*}$ (1.9±0.1$^{d*}$) | 0.8±0.0$^{c}$ (0.3±0.1$^{a*}$) |
| *aap* | EAEC 042 | 0.0±0.0$^{a}$ (0.0±0.0$^{a*}$) | 0.0±0.0$^{a*}$ (0.0±0.0$^{a*}$) | 0.0±0.0$^{a*}$ (0.0±0.0$^{a*}$) | 0.0±0.0$^{a*}$ (0.02±0.0$^{a*}$) | 0.0±0.0$^{a*}$ (0.0±0.0$^{a*}$) |
| | EAHEC O104:H4 | 0.7±0.2$^{b*}$ (2.0±0.0$^{d*}$) | 4.2±0.1$^{e*}$ (1.0±0.0$^{c}$) | 0.2±0.1$^{a*}$ (1.9±0.0$^{d*}$) | 2.6±0.2$^{d*}$ (3.0±0.0$^{e*}$) | 0.2±0.0$^{a*}$ (1.0±0.0$^{c}$) |
| *stx2a* | EAHEC O104:H4 | 0.4±0.1$^{b*}$ (1.2±0.2$^{d*}$) | 0.5±0.0$^{b*}$ (0.1±0.0$^{a*}$) | 0.9±0.1$^{c}$ (0.9±0.0$^{c}$) | 1.3±0.1$^{c}$ (5.9±0.2$^{e*}$) | 0.7±0.0$^{b*}$ (1.1±0.2$^{c}$) |
| | EHEC O157:H7 | 0.5±0.1$^{b*}$ (0.5±0.2$^{b*}$) | 1.2±0.0$^{c}$ (0.6±0.1$^{b*}$) | 0.4±0.1$^{b*}$ (0.4±0.1$^{b*}$) | 2.6±0.2$^{d*}$ (3.2±0.3$^{e*}$) | 0.7±0.1$^{b*}$ (2.1±0.2$^{d*}$) |
| *eae* | EHEC O157:H7 | 3.9±0.0$^{e*}$ (4.1±0.5$^{e*}$) | 2.3±0.0$^{d*}$ (0.1±0.3$^{a*}$) | 0.8±0.0$^{c}$ (0.1±0.0$^{a*}$) | 5.7±0.5$^{e*}$ (4.1±0.3$^{e*}$) | 0.6±0.4$^{b}$ (0.8±0.0$^{1c}$) |

The header spanning columns: "Fold change in expression in adhered (or non-adhered) bacteria".

Bacteria were exposed to antimicrobials for 1 h and aliquots (containing bacteria and the antimicrobial) were then added to confluent monolayers of HEp-2 cells for 3 h.
±: Standard deviation. 0.0 = ≤0.04 (for standard deviation and fold change)
Different letters indicate significant differences from the control. Adhered and non-adhered bacteria were analyzed separately.
The "primary" control group (adhered and non-adhered bacteria after adhesion assay without antimicrobials) was normalized as 1; >1 indicates gene upregulation and <1 indicates gene downregulation.
* Significant difference ($p \leq 0.05$)

*eae* (intimin) was upregulated by rifaximin and 0.40 mg/ml of oregano extract but was unaltered by carvacrol (0.010 mg/ml; Table 1).

We then evaluated whether the fitness of the HEp-2 cells during the adhesion assay was affected by the presence of the bacteria and antimicrobials together or separately. To this end, we evaluated the expression of oxidative stress-related genes (superoxide dismutase [cytoplasmic *SOD1* and mitochondrial *SOD2*], catalase [*CAT*], and glutathione peroxidase [*GPx*]) in HEp-2 cells after antimicrobial treatment with/without EAHEC/EAEC. *SOD1* was upregulated by rifaximin, carvacrol, and oregano extract up to 1.1-fold in all cases involving EAHEC (Table 2). *SOD2* exhibited a similar trend in the presence of the highest concentrations of rifaximin and oregano extract. Regarding the EAEC-treated HEp-2 cells, the natural antimicrobials led to up to 1.6-fold *SOD1* upregulation, except 0.40 mg/ml oregano extract (which led to no change). Additionally, 0.005 mg/ml rifaximin upregulated *SOD2* by up to 2.9-fold. *CAT* and *GPx* in the HEp-2 cells were downregulated by treatment with either EAEC or EAHEC plus antimicrobials by at least 0.5-fold (p≤0.05). The effects on *SOD1*, *SOD2*, *CAT*, and *GPx* were lower (or even not seen) when only the antimicrobials were present (fold changes: 0.6–1.6). Thus, the changes in the expression of these genes are a response to the presence of bacteria and/or antimicrobials.

In conclusion, antimicrobials at sub-MBC alter the adherence of *E. coli* to HEp-2 cells by changing the expression of *E. coli* adherence-related genes, and carvacrol also disrupted the aggregative pattern of EAHEC. The expression of oxidative stress-related genes in HEp-2 cells was also affected by treatment with the bacteria plus antimicrobials.

## Exposure of *E. coli* to antimicrobials is sufficient to decrease their adhesion to HEp-2 cells

We found that preincubation of *E. coli* with antimicrobials changed the adhesion to HEp-2 cells, and we wondered whether the antimicrobial concentrations in the bacterial culture

**Table 2. Effects of antimicrobials on the fold changes in the expression of oxidative stress-related genes in HEp-2 cells.**

| Gene | Treatment of HEp-2 cells | Control groups (HEp-2 cells + bacteria) | Fold change in expression in adhered (or non-adhered) bacteria | | | | |
|---|---|---|---|---|---|---|---|
| | | | Rifaximin (mg/ml) | | Carvacrol (mg/ml) | Oregano extract (mg/ml) | |
| | | | 0.005 | 0.002 | 0.010 | 0.40 | 0.20 |
| *SOD1* | Only AM | | 1.1±0.1[c] | 0.9±0.0[c] | 1.1±0.2[c] | 0.5±0.1[b*] | 1.6±0.1[d*] |
| | AM + EAEC | 0.8±0.0[c] | 2.6±0.0[e*] | 1.5±0.2[d*] | 1.2±0.1[c] | 0.9±0.1[c] | 1.6±0.1[d*] |
| | AM + EAHEC | 1.5±0.0[d*] | 2.1±0.0[e*] | 1.4±0.1[d*] | 1.2±0.0[c] | 1.8±0.0[d*] | 1.3±0.0[d*] |
| *SOD2* | Only AM | | 1.1±0.1[c] | 0.7±0.1[c] | 1.2±0.0[c] | 0.7±0.0[c] | 1.4±0.1[d*] |
| | AM + EAEC | 0.3±0.0[a*] | 2.9±0.0[e*] | 1.4±0.1[d*] | 0.3±0.0[a*] | 0.6±0.2[b*] | 1.1±0.0[c] |
| | AM + EAHEC | 1.0±0.0[c] | 2.8±0.0[e*] | 0.8±0.1[c] | 0.8±0.1[c] | 1.4±0.1[d*] | 0.5±0.1[a*] |
| *CAT* | Only AM | | 0.7±0.0[b*] | 1.0±0.1[c] | 0.7±0.0[b*] | 1.5±0.2[d*] | 1.0±0.0[c] |
| | AM + EAEC | 0.0±0.0[a*] | 0.3±0.1[a*] | 0.1±0.0[a*] | 0.1±0.0[a*] | 0.1±0.1[a*] | 0.1±0.0[a*] |
| | AM + EAHEC | 0.1±0.0[a*] | 0.1±0.1[a*] | 0.1±0.0[a*] | 0.0±0.0[a*] | 0.1±0.0[a*] | 0.1±0.0[a*] |
| *GPx* | Only AM | | 0.8±0.0[c] | 1.1±0.1[c] | 0.9±0.1[c] | 1.0±0.0[c] | 0.6±0.1[b*] |
| | AM + EAEC | 1.7±0.0[d*] | 0.4±0.0[a*] | 0.3±0.1[a*] | 0.5±0.0[b*] | 0.5±0.0[b*] | 0.0±0.0[a*] |
| | AM + EAHEC | 0.4±0.0[b*] | 0.3±0.0[a*] | 0.5±0.0[b*] | 0.4±0.0[a*] | 0.3±0.1[a*] | 0.4±0.1[b*] |

Bacteria were exposed to antimicrobials for 1 h and then added to confluent HEp-2 cells for 3 h before analysis of the HEp-2 cells.

AM: antimicrobial; *SOD1*: superoxide dismutase 1; *SOD2*: superoxide dismutase 2; *CAT*: catalase, *GPx*: glutathione peroxidase; EAEC: *E. coli* 042, EAHEC: *E. coli* O104:H4

±: Standard deviation. 0.0 = ≤0.04 (for standard deviation and fold change)

Different letters indicate significant differences from the control. Adhered and non-adhered bacteria were analyzed separately. The "primary" control group (HEp-2 cells with no bacteria and no AM) was normalized as 1; >1 indicates gene upregulation and <1 indicates gene downregulation.

* Significant difference ($p \leq 0.05$)

indirectly played a role in this effect by affecting the HEp-2 cells. Thus, we preincubated *E. coli* with the antimicrobials for 1 h and, instead of directly inoculating the HEp-2 cells, we washed the bacteria to remove the medium and antimicrobial and then added the washed bacteria to the HEp-2 cells (S1B Fig). We found that bacterial adhesion decreased, but the effect was modest (up to 23.7% with 0.025 mg/ml carvacrol, Table 3) compared to that when, along with the bacteria, small amounts of antimicrobials were added to the HEp-2 cells (64.5%, Fig 1). Importantly, Hb extract and brazilin again caused hyper-adherence (with more adherence than in the control group), as in the earlier adhesion assays (Table 3). Because of this, brazilin and Hb extract were not used in the remainder of the assays. The results also showed that the culture medium in which bacteria were grown (EMEM or MH broth) did not influence the adhesion ability (Table 3). Overall, the results suggest that preincubation of *E. coli* with antimicrobials, without exposing the Hep-2 cells to the antimicrobials, is sufficient to impact their adhesion ability.

## Antimicrobials do not protect HEp-2 cells against *E. coli*

As the antimicrobials reduced *E. coli* adherence to HEp-2 cells, we wanted to determine whether incubation of HEp-2 cells with the antimicrobials would protect against bacterial adhesion. To explore this, HEp-2 cells were incubated with antimicrobials at sub-MBC for 1 h, and then challenged with the bacteria. Almost in general, we found that preincubation of the HEp-2 cells with antimicrobials did not influence (p ≤ 0.05) the adherence of any of the *E. coli* strains (Table 4), with exception of rifaximin (0.002 mg/ml) that caused hyper-adherence (up to 105.8%, p≤0.05) in EAEC and EHEC, and carvacrol where a reduction of adherence (2.8%, p≤0.05) was detected in EAHEC. Therefore, the decreased adherence of *E. coli* to HEp-2 cells was primarily due to a reduction of the binding ability of *E. coli*, rather than protecting HEp-2 cells against adhesion.

**Table 3. Effects of antimicrobials on percentage of *E. coli* adhered to HEp-2 cells after preincubating the bacteria with antimicrobials.**

| *E. coli* serotype | log$_{10}$ bacteria adhered to HEp-2 cells (%) | | | | | | | | | | |
|---|---|---|---|---|---|---|---|---|---|---|---|
| | | Antimicrobial (mg/ml) | | | | | | | | | |
| | Control | Rifaximin | | Carvacrol | | Oregano extract | | Brazilin | | Hb extract | |
| | | 0.005 | 0.002 | 0.025 | 0.010 | 0.40 | 0.20 | 1.5 | 1.0 | 3.0 | 1.5 |
| | *E. coli* growing in EMEM | | | | | | | | | | |
| EAHEC O104:H4 | 8.7±0.2 (100)[d] | 6.9±0.1 (78.9)[a*] | 7.9±0.1 (90.7)[c*] | 7.1±0.3 (80.2)[ab*] | 7.7±0.4 (88.7)[c*] | 7.3±0.3 (83.5)[ab*] | 7.5±0.6 (85.8)[bc*] | 8.9±0.6 (101.9)[d] | 9.1±0.5 (103.9)[d] | 8.9±0.52 (102.0)[d] | 9.1±0.1 (104.1)[d] |
| EAEC 042 Chile | 7.4±0.1 (100)[e] | 5.7±0.1 (76.9)[a*] | 6.6±0.4 (88.7)[bc*] | 5.8±0.2 (77.6)[a*] | 6.5±0.2 (87.8)[b*] | 6.8±0.1 (91.9)[c*] | 6.8±0.1 (92.3)[c*] | 7.8±0.4 (105.1)[f*] | 7.4±0.3 (100.3)[e] | 7.6±0.3 (102.8)[ef] | 7.5±0.5 (101.3)[ef] |
| EHEC O157:H7 | 8.0±0.6 (100)[d] | 7.3±0.5 (90.7)[c*] | 8.01±0.5 (99.8)[d] | 6.1±0.6 (76.5)[a*] | 6.6±0.6 (84.3)[b*] | 6.6±0.4 (82.9)[b*] | 6.8±0.7 (85.1)[b*] | 8.1±0.6 (101.5)[d] | 8.4±0.4 (104.7)[e*] | 8.0±0.5 (100.9)[d] | 8.2±0.2 (101.7)[d] |
| | *E. coli* growing in MH broth | | | | | | | | | | |
| EAHEC O104:H4 | 8.8±0.2 (100)[d] | 6.8±0.2 (78.1)[a*] | 7.76±0.2 (90.2)[c*] | 6.9±0.1 (79.6)[a*] | 7.8±0.4 (89.4)[c*] | 7.3±0.09 (83.3)[ab*] | 7.5±0.51 (86.5)[bc*] | 8.8±0.3 (100.7)[d] | 9.0±0.3 (102.6)[d] | 8.9±0.5 (101.5)[d] | 9.1±0.4 (104.0)[d] |
| EAEC 042 Chile | 7.6±0.3 (100)[e] | 5.8±0.3 (78.7)[a*] | 6.56±0.2 (88.5)[b*] | 5.8±0.2 (77.7)[a*] | 6.6±0.2 (89.4)[bc*] | 6.9±0.3 (92.4)[cd*] | 7.0±0.1 (94.8)[d*] | 7.8±0.2 (105.2)[f] | 7.7±0.1 (103.6)[f] | 7.6±0.3 (102.1)[ef] | 7.7±0.2 (104.01)[f] |
| EHEC O157:H7 | 7.9±0.5 (100)[d] | 7.2±0.5 (90.2)[c*] | 8.2±0.5 (101.9)[d] | 6.2±0.4 (77.5)[a*] | 6.6±0.5 (82.0)[ab*] | 6.6±0.3 (82.7)[ab*] | 6.7±0.3 (83.7)[b*] | 8.1±0.2 (101.9)d | 8.3±0.4 (103.0)[d] | 8.1±0.4 (101.7)[d] | 8.0±0.2) (100.2)[d] |

Bacteria were grown in EMEM or Mueller Hinton (MH) broth and preincubated with antimicrobials for 1 h, the medium was discarded, and the bacteria were resuspended in fresh EMEM and then used to infect HEp-2 monolayers for 3 h before assessing the adhered *E. coli* cells.

±: Standard deviation.

Different letters indicate significant differences from the control. Each strain was analyzed individually. The "primary" control group (adhered bacteria to HEp-2 cells after finishing the method previously described without antimicrobials) that was normalized as 100, <100 indicates inhibition of adhesion and >100 indicates hyper-adherence.

* Significant difference ($p \leq 0.05$)

## Antimicrobials cannot detach EAEC, EHEC, or EAHEC already adhered to HEp-2 cells

Next, we assessed whether the antimicrobials can disrupt an already established adherence of *E. coli* to HEp-2 cells (S1D Fig) for 1 h. Once the bacteria–HEp-2 cell adhesion was established, the medium containing the non-adhered bacteria was removed, fresh EMEM with the antimicrobial at sub-MBC was added for

**Table 4. Percentage of *E. coli* adhered to HEp-2 cells after preincubating the HEp-2 cells with antimicrobials.**

| *E. coli* serotype | log$_{10}$ bacteria adhered to HEp-2 cells (%) | | | | | |
|---|---|---|---|---|---|---|
| | | Antimicrobial (mg/ml) | | | | |
| | Control | Rifaximin | | Carvacrol | Oregano extract | |
| | | 0.005 | 0.002 | 0.010 | 0.40 | 0.20 |
| EAHEC O104:H4 | 8.9±0.1 (100)[b] | 8.9±0.1 (99.7)[b] | 9.0±0.3 (100.6)[bc] | 8.7±0.1 (97.2)[a*] | 8.9±0.1 (99.5)[b] | 9.0±0.3 (100.3)[bc] |
| EAEC 042 Chile | 7.7±0.1 (100)[a] | 7.8±0.2 (101.6)[abc] | 8.0±0.4 (103.9)[d*] | 7.6±0.1 (99.6)[a] | 7.9±0.1 (102.4)[bcd] | 7.9±0.1 (102.6)[bcd] |
| EHEC O157:H7 | 8.1±0.1 (100)[bc] | 7.9±0.1 (98.0)[a] | 8.5±0.1 (105.8)[d*] | 8.1±0.1 (100.0)[bc] | 8.0±0.3 (99.3)[ab] | 8.2±0.1 (101.4)[c] |

Confluent monolayers of HEp-2 cells were preincubated with antimicrobials for 1 h and bacteria were then added for 3 h before assessing the adhered E. coli.

±: Standard deviation.

Different letters indicate significant differences from the control. Each strain was analyzed individually. The "primary" control group (adhered bacteria to HEp-2 cells after finishing the method previously described without antimicrobials) that was normalized as 100, <100 indicates inhibition of adhesion and >100 indicates hyper-adherence.

* Significant difference ($p \leq 0.05$) compared with control

**Table 5. Effects of antimicrobials at sub-MBC on removal of *E. coli* already adhered to HEp-2 cells.**

| *E. coli* serotype | Control | Antimicrobial (mg/ml) | | | | |
|---|---|---|---|---|---|---|
| | | Rifaximin | | Carvacrol | Oregano extract | |
| | | 0.005 | 0.002 | 0.010 | 0.40 | 0.20 |
| EAHEC O104:H4 | 7.4±0.3 (100)[b] | 7.0±0.1 (95.3)[ab] | 7.2±0.1 (97.5)[ab] | 7.3±0.1 (99.5)[b] | 7.1±0.1 (96.7)[ab] | 7.3±0.2 (99.8)[b] |
| EAEC 042 Chile | 6.1±0.1 (100)[a] | 6.0±0.3 (98.6)[a] | 6.0±0.1 (99.4)[a] | 6.0±0.2 (98.7)[a] | 6.1±0.1 (100.8)[a] | 6.1±0.1 (101.4)[a] |
| EHEC O157:H7 | 6.7±0.1 (100)[a] | 6.45±0.4 (96.4)[a] | 6.5±0.3 (97.1)[a] | 6.7±0.4 (99.4)[a] | 6.5±0.3 (97.6)[a] | 6.5±0.3 (97.3)[a] |

Bacteria were used to infect HEp-2 monolayers for 1 h and then antimicrobials were added for 3 h and the adhered bacteria were measured.

±: Standard deviation.

Similar letters indicate similar behavior to control. The "primary" control group (adhered bacteria to HEp-2 cells after finishing the method previously described without antimicrobials) that was normalized as 100% <100 indicates inhibition of adhesion and >100 indicates hyper-adherence.

Each strain was analyzed individually.

3 h, and then the number of adhered bacteria was determined. The results indicate that the antimicrobials had no effect (p≤0.05) on already established bacteria–HEp-2 cell adhesion (Table 5).

## Decreased adhesion ability exhibited by antimicrobial-treated EAEC to HEp-2 cells lasts several hours

Having established that the antimicrobials affect the initial adherence of *E. coli* to HEp-2 cells (Figs 1 and 2), and that this effect is probably due to changes in *E. coli* gene expression (Table 1), we wondered whether the effects could last for several hours (S1E Fig). Thus, the non-adhered bacteria from the first adhesion assay (previously exposed to antimicrobials) were washed and added to HEp-2 cells for 3 h, and this process was repeated in second and third cycles. In each cycle, we determined the number of adhered bacteria.

We found that the adhesion ability recovered gradually with each cycle. Carvacrol (0.10 mg/ml) inhibited EAHEC adhesion by 54.2%, 19.8%, and 7.0% in the first, second, and third cycle, respectively. Similar trends were observed regarding oregano extract and EAEC adhesion. In contrast, EHEC rapidly recovered their adherence ability, with at least 95.8% of bacteria adhered in the first cycle for all antimicrobials (Table 6).

The adhesion-related gene expression in the initial non-adhered bacteria showed inconsistent patterns (Table 1). Only *aggA* in the two aggregative strains (EAEC 042 and EAHEC O104:H4) showed a consistent change, being downregulated in most of the treatments (Table 1). This downregulation of *aggA* (which encodes a major component of the AAF) would induce alterations in the AAF, which is involved in bacterial adhesion. However, this downregulation was generally only observed in the unadhered cells, rather than in the adhered cells analyzed above.

The *aap* gene (which encodes dispersin) was downregulated in EAEC in all cases, and in EAHEC, it was upregulated by carvacrol and by rifaximin and oregano (higher sub-CMB concentrations), whereas in the rest of treatments was unaltered (Table 1). Interestingly, adhered bacteria seemed to behave quite different than non-adhered, as there was higher gene upregulation in adhered bacteria in many cases, with the exception of *eae* (which encodes intimin in EHEC), which exhibited similar expression in the adhered and non-adhered bacteria in most cases (with exception of lower sub-CMB of rifaximin and oregano extract, Table 1).

Overall, the antimicrobials had profound effects on bacterial adhesion that lasted several hours. There were questions regarding whether the non-adherent affected bacteria could

**Table 6. Effects of antimicrobials on the later adherence of initially non-adhered *E. coli*.**

| *E. coli* serotype | Cycle | Control | $\log_{10}$ bacteria adhered to HEp-2 cells (%) | | | | |
| --- | --- | --- | --- | --- | --- | --- | --- |
| | | | Rifaximin | | Carvacrol | Oregano extract | |
| | | | 0.005 | 0.002 | 0.010 | 0.40 | 0.20 |
| EAHEC O104:H4 | 0 | 8.3±0.2 (100)[e] | 4.9±0.1 (56.3)[c*] | 5.4±0.3 (62.1)[d*] | 3.1±0.2 (35.5)[a*] | 4.2±0.1 (48.90)[b*] | 4.7±0.3 (53.7)[c*] |
| | 1st | 8.7±0.1 (100)[f] | 7.2±0.3 (82.2)[d*] | 8.2±0.5 (93.8)[e*] | 4.0±0.1 (45.8)[a*] | 5.2±0.1 (59.5)[b*] | 5.8±0.3 (66.5)[c*] |
| | 2nd | 8.7±0.4 (100)[b] | 9.6±0.1 (110.2)c* | 9.4±0.1 (108.3)[c*] | 7.0±0.1 (80.2)[a*] | 8.4±0.4 (96.6)[b] | 8.5±0.4 (97.3)[b] |
| | 3rd | 8.9±0.3 (100)[b] | 9.1±0.1 (105.2)[bc] | 9.01±0.5 (106.5)[c*] | 8.1±0.1 (93.0)[a*] | 8.5±0.3 (98.0)[b] | 8.5±0.3 (99.7)[b] |
| EAEC 042 Chile | 0 | 7.6±0.3 (100)[e] | 2.9±0.1 (38.6)[a*] | 4.4±0.4 (58.1)[d*] | 3.0±0.1 (39.8)[ab*] | 3.3±0.1 (43.1)[b*] | 3.9±0.4 (52.3)[c*] |
| | 1st | 7.4±0.2 (100)[f] | 4.9±0.1 (65.1)[c*] | 6.8±0.3 (89.5)[e*] | 4.3±0.3 (56.9)[a*] | 4.3±0.2 (61.5)[b*] | 5.3±0.1 (69.8)[d*] |
| | 2nd | 7.5±0.2 (100)[b] | 7.4±0.2 (98.2)[b] | 7.9±0.1 (105.1)[bc] | 6.3±0.4 (84.1)[a*] | 7.5±0.1 (98.7)[b] | 7.4±0.1 (97.7)[b] |
| | 3rd | 7.5±0.3 (100)[ab] | 7.7±0.2 (102.3)[b] | 7.6±0.2 (100.8)[b] | 7.3±0.1 (96.3)[a] | 7.5±0.1 (99.2)[ab] | 7.4±0.3 (98.2)[ab] |
| EHEC O157:H7 | 0 | 7.9±0.2 (100)[e] | 5.0±0.1 (62.9)[c*] | 5.8±0.1 (74.0)[d*] | 3.8±0.2 (48.1)[a*] | 4.4±0.3 (55.4)[b*] | 5.0±0.2 (63.2)[c*] |
| | 1st | 7.9±0.1 (100)[bc] | 7.7±0.1 (97.6)[ab] | 8.0±0.2 (101.4)[bc] | 7.6±0.2 (95.8)[a*] | 7.8±0.2 (98.5)[b] | 7.9±0.1 (99.8)[bc] |
| | 2nd | 8.0±0.1 (100)[a] | 8.2±0.2 (104.0)[b*] | 8.4±0.2 (106.1)[b*] | 7.9±0.2 (99.6)[a] | 7.9±0.4 (99.7)[a] | 7.9±0.3 (100.6)[a] |
| | 3rd | 8.0±0.1 (100)[a] | 8.0±0.1 (101.7)[a] | 8.0±0.4 (103.0)[a] | 8.0±0.1 (100.1)[a] | 7.8±0.2 (99.7)[a] | 7.9±0.1 (98.8)[a] |

*E. coli* non-adhered to HEp-2 cells after antimicrobial exposure were used to infect HEp-2 monolayers in three consecutive cycles and the number of adhered *E. coli* cells in each cycle was determined by plate count.

±: Standard deviation.

Different letters indicate significant differences from the control. Each strain was analyzed individually by cycle. The "primary" control group (adhered bacteria to HEp-2 cells after finishing the method previously described on each cycle without antimicrobials) that was normalized as 100%, <100 indicates inhibition of adhesion and >100 indicates hyper-adherence.

* Significant difference ($p < 0.05$)

maintain their adhesion ability, whether the effect is maintained as a phenotypic trait, and whether the effect is induced by gene expression. It seems that the effect is derived from changes in gene expression. As generations of bacteria arose across the cycles of adhesion assays, the expression of adhesion-related genes gradually recovered, and so did the adhesion ability (Table 6). EAHEC recovered gradually after exposure to both rifaximin concentrations, even inducing hyper-adherence (up to 110.2%) in the second and third cycles.

## Discussion

Pathogenic bacteria express molecules or form structures that promote adhesion to host cells, which is a critical process before colonization, toxin secretion, and/or induction of host cell responses. Thus, adhesion is a key event of bacterial pathogenesis to be studied [28], and it represents a potential target for controlling pathogenic bacteria. Therefore, the effect of antimicrobials on the adhesion ability of pathogenic *E. coli* strains was studied. The antimicrobials showed bactericidal effects against the studied strains; the highest MBCs corresponded to crude oregano and Hb extracts, which also showed greater variability between strains. The corresponding major compounds, carvacrol and brazilin, respectively, had lower MBCs, which concurs with previous research [29].

The mechanisms of action of the antimicrobials studied are based on bacterial membrane depolarization; however, at the concentrations used, the membrane integrity remained stable. Carvacrol (a major compound of oregano) targets membrane-associated proteins, damaging bacterial cytoplasmic membranes [30, 31]. Oregano extract has also shown activity against the bacterial membrane, but its mechanism of action is broader and less specific because the extract is composed of several active molecules [32]. Hb extract also contains various

compounds, such as brazilin and hematoxylin [33]. Brazilin exerts its antibacterial effects by inhibiting the protein synthesis in bacteria [34, 35].

Rifaximin is a commercial synthetic antibiotic that exerts its effects by binding to RNA polymerase, preventing DNA synthesis [36]. This antibiotic was the recommended therapy during the *E. coli* O104:H4 outbreak in 2011. However, in retrospective analyses of Shiga toxin-producing *E. coli* O157:H7 outbreaks and sporadic infections, patients treated with antibiotics were found to have an increased risk of hemolytic uremic syndrome [37]. In theory, antibiotic treatment for EHEC infections may lead to higher toxicity due to a massive release of Shiga toxin after bacterial death during the prodromal phase of diarrhea [17].

An important aspect was to check that the antimicrobials were not toxic to HEp-2 cells at the concentrations selected. Carvacrol was toxic to HEp-2 cells at 0.025 mg/ml (so only 0.010 mg/ml was used in the subsequent assays), concurring with research showing that carvacrol at 0.2–0.4 mg/ml decreases neuroblastoma N2a cell proliferation [38], although the HEp-2 cells were even more sensitive in our assays.

In general, the adhesion assays showed that rifaximin, carvacrol, and oregano extract decreased ($p \leq 0.05$) the ability of *E. coli* strains to adhere to the HEp-2 cells. As expected, reduced exposure duration of the bacteria to the antimicrobials (e.g., 1 h), lessened the change in the adhesion percentage. Additionally, the presence of antimicrobials in the initial phase of adhesion (e.g., 1 h of preincubation plus 3 h of the adhesion test) caused the highest reductions. Carvacrol at subinhibitory concentrations has been reported to reduce the adhesiveness of *E. coli* to vaginal cells and the invasion of porcine intestinal epithelial cells (IPEC-J2) by *Salmonella* [39, 40].

Brazilin and Hb extract had the opposite effect, causing increased adhesion. This effect could be due to the potential of these antimicrobials to inhibit telomerase, which has been reported to lead to anticancer effects [41].

Although exposure of HEp-2 cells to antimicrobials at sub-MBC did not affect the subsequent bacterial adhesion in this study (i.e., antimicrobials do not protect HEp-2 cells against *E. coli*), Brown *et al* [42] reported that rifaximin reduced the number of adhered bacteria. The difference could be due to variations in the concentrations used and the exposure duration; <100 μmol/l rifaximin is not cytotoxic to epithelial cells (viability >80%), but higher concentrations can cause changes in epithelial cell physiology [43].

The adherence process in EAEC is related to the proteins encoded by the pAA plasmid; it requires fimbriae structures (AAF) and surface proteins (dispersin). The master regulator that controls the expression of the genes that encode these proteins is *aggR* [44]. As phenotypic responses require molecular changes, we analyzed several adhesion-related genes to try to elucidate the mechanisms of action of the antimicrobials against the bacteria; however, firm conclusions could not be reached as at least one of the four investigated genes in EAEC (*aggR*, *aggA*, *aap* or *pic*) was downregulated.

Changes in the expression of *aap* (which encodes dispersin, which is responsible for the anti-aggregation phenotype) alter the membrane surface. A previous study showed that inhibition of dispersin alters the outer membrane polarization, which in turn affects the electrostatic interactions involving the AAF [45]. In our study, rifaximin and oregano extract downregulated *aap* in EAHEC strain, which could explain why the bacteria, while also exhibiting decreased adhesion, exhibited a tendency to aggregate (due to lack of dispersin) after rifaximin and oregano extract treatment. However, despite the *aap* downregulation, the marked anti-aggregation effect of carvacrol on EAHEC suggested that carvacrol somehow promotes an anti-aggregation phenotype (e.g., by promoting the anti-aggregating effect possibly due to the disproportion between AAFs protein and dispersin expressed). A low amount of dispersin increases the negative charge on the membrane, which affects the separation of the AAFs. AAF

fimbriae are highly hydrophobic, which favors the strong self-agglutination of bacteria in an aqueous environment. However, the absence of dispersin results in notable alterations in the morphology of AAF fimbriae, which adhere to the surface of the bacterial cell [46], which could explain the non-aggregation behavior in the strain EAHEC.

Interestingly, the size of the major protein component of the AAF encoded by *aggA* differs between the two aggregative strains, EAEC and EAHEC, which may have caused the difference in the characteristic stacked brick pattern between EAEC and EAHEC, as the pattern was more altered for EAHEC. More precisely, the *aggA* gene is 483 bp and encodes a 160-amino acid protein in EAEC 042 (GenBank accession number: AB571092.1: 34–516), while it is 351 bp and encodes a smaller 116-amino acid protein in EAHEC O104:H4 (GenBank accession number: AFOB02000132.1; NCBI, Bethesda, MD, USA). This smaller AAF-related protein, coupled with downregulated *aap* (dispersin) in EAHEC treated with 0.010 mg/ml of carvacrol, may have led to the phenotypic change in EAHEC, independently of the reduced adhesion observed in the aggregative strains.

Importantly, oregano extract (0.40 mg/ml) slightly upregulated (4.9-fold ($p \leq 0.05$) *stx2a* (harbored by the Stx2a phage) in non-adhered EAHEC; this effect was also observed on the adhered EHEC and EAHEC bacteria (upregulated by 1.6-fold [$p \leq 0.05$] and 0.3-fold, respectively). The *Stx2a* gene was downregulated in the adhered bacteria by the other antimicrobial tested. The use of antibiotics for Shiga toxin-producing *E. coli* O157:H7 infections remains controversial due to concern about triggering hemolytic-uremic syndrome by causing Stx release during treatment. Bielaszewska et al. [21] reported that rifaximin did not upregulate *stx2* harbored by phages in EAHEC O104:H4, whereas [47] reported that sub-lethal concentrations of rifaximin upregulated *stx2* in EHEC O157:H7, indicating that the effects may depend on the rifaximin dose. The downregulation of *stx2a* in EHEC and EAEHC treated with carvacrol observed in this study is in accordance with the results reported by Mith *et al* [48], who suggested that carvacrol is a potent mitigator of adverse health effects caused by virulence gene expression in EAHEC O104:H4. However, it is important to note that the essential oil is usually used rather than the ethanolic extract used in this study. Although both the ethanolic extract and the essential oil exhibit antibacterial properties, the highest activity has been reported for the essential oil. The activity observed with the essential oil could be related to the presence of carvacrol and thymol as major compounds, while the oregano extract contains a larger variety of compounds (limonene, gamma-cariofilene, rhocymenene, canfor, linalool, alpha-pinene, carvacrol, and thymol) [49, 50].

The addition of the antimicrobials after *E. coli*–HEp-2 adhesion was not effective in separating the bacteria and HEp-2 cells. However, this finding allows us to highlight the importance of the timepoint at which the target cells are exposed to the antimicrobials.

In summary, we found that rifaximin, carvacrol, and oregano extract decrease the ability of all the strains under study to adhere to HEp-2 cells when they are added at the initial stage of the adhesion process. In contrast, brazilin and Hb extract increased the adherence. Carvacrol modified the adhesion pattern of EAHEC, and the antimicrobials induced dose-dependent expression changes in adhesion-related genes. This study shows that these antimicrobials may have important roles in preventing aggregative and hemorrhagic *E. coli* infections.

## Supporting information

**S1 Fig. Efficacy of subinhibitory concentrations of compounds/extracts to inhibit *E. coli* adhesion.**
(DOCX)

**S1 Table. Genes and primers used to detect the expression of virulence genes of *E. coli* strains and the expression of oxidative stress genes in HEp-2 cells, after adhesion assay.**
(DOCX)

**S2 Table. Minimal bactericidal concentration (MBC, mg/ml) of natural extracts and compounds of three E. coli serotypes.** Values in parentheses indicate sub-MBC concentrations used.
(DOCX)

**S3 Table. pH measurement of culture media with antimicrobial compounds.**
(DOCX)

**S4 Table. Percentage of bacterial mortality after the exposure to higher sub-CMB of antimicrobials (determined by flow cytometry).**
(DOCX)

**S5 Table. Percentage of HEp-2 cells viability by MTT method determination after exposure to sub-CMB of antimicrobials (mg/ml).**
(DOCX)

**S6 Table. Number of bacteria and bacterial aggregates adhered to HEp-2 cells stained by GIEMSA.**
(DOCX)

## Author Contributions

**Conceptualization:** Santos García, Norma Heredia.

**Data curation:** Yaraymi Ortiz, Alam García-Heredia.

**Formal analysis:** Yaraymi Ortiz.

**Funding acquisition:** Norma Heredia.

**Investigation:** Santos García.

**Methodology:** Yaraymi Ortiz, Alam García-Heredia, Angel Merino-Mascorro, Santos García, Norma Heredia.

**Project administration:** Norma Heredia.

**Resources:** Norma Heredia.

**Supervision:** Alam García-Heredia, Angel Merino-Mascorro, Santos García.

**Validation:** Yaraymi Ortiz, Alam García-Heredia, Angel Merino-Mascorro.

**Visualization:** Yaraymi Ortiz, Luisa Solís-Soto.

**Writing – original draft:** Yaraymi Ortiz, Alam García-Heredia, Norma Heredia.

**Writing – review & editing:** Yaraymi Ortiz, Alam García-Heredia, Angel Merino-Mascorro, Santos García, Luisa Solís-Soto, Norma Heredia.

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
