## [Decision Letter · Decision Letter 0]

22 Dec 2020

PONE-D-20-36066

Natural and synthetic antimicrobials reduce adherence of enteroaggregative and enterohemorrhagic Escherichia coli to epithelial cells

PLOS ONE

Dear Dr. Heredia,

Thank you for submitting your manuscript to PLOS ONE. After careful consideration, we feel that it has merit but does not fully meet PLOS ONE’s publication criteria as it currently stands. Therefore, we invite you to submit a revised version of the manuscript that addresses the points raised during the review process.Your manuscript has been reviewed by an expert in your field and a minor revision is suggested. 

Please submit your revised manuscript by two weeks. If you will need more time than this to complete your revisions, please reply to this message or contact the journal office at plosone@plos.org. Please include the following items when submitting your revised manuscript:

We look forward to receiving your revised manuscript.

Kind regards,

Yung-Fu Chang

Academic Editor

PLOS ONE

Journal Requirements:

2.) We note that you have included the phrase “data not shown” in your manuscript. Unfortunately, this does not meet our data sharing requirements. PLOS does not permit references to inaccessible data. We require that authors provide all relevant data within the paper, Supporting Information files, or in an acceptable, public repository. Please add a citation to support this phrase or upload the data that corresponds with these findings to a stable repository (such as Figshare or Dryad) and provide and URLs, DOIs, or accession numbers that may be used to access these data. Or, if the data are not a core part of the research being presented in your study, we ask that you remove the phrase that refers to these data.

3.)PLOS ONE now requires that authors provide the original uncropped and unadjusted images underlying all blot or gel results reported in a submission’s figures or Supporting Information files. This policy and the journal’s other requirements for blot/gel reporting and figure preparation are described in detail at https://journals.plos.org/plosone/s/figures#loc-blot-and-gel-reporting-requirements and https://journals.plos.org/plosone/s/figures#loc-preparing-figures-from-image-files. When you submit your revised manuscript, please ensure that your figures adhere fully to these guidelines and provide the original underlying images for all blot or gel data reported in your submission. See the following link for instructions on providing the original image data: https://journals.plos.org/plosone/s/figures#loc-original-images-for-blots-and-gels.

4.) Please upload a new copy of Figure 1 as the detail is not clear. Please follow the link for more information: https://blogs.plos.org/plos/2019/06/looking-good-tips-for-creating-your-plos-figures-graphics/" https://blogs.plos.org/plos/2019/06/looking-good-tips-for-creating-your-plos-figures-graphics/

Reviewers' comments:

Reviewer's Responses to Questions

**Comments to the Author**

1. Is the manuscript technically sound, and do the data support the conclusions?

Reviewer #1: Yes

2. Has the statistical analysis been performed appropriately and rigorously? 

Reviewer #1: Yes

3. Have the authors made all data underlying the findings in their manuscript fully available?

Reviewer #1: Yes

4. Is the manuscript presented in an intelligible fashion and written in standard English?

Reviewer #1: Yes

5. Review Comments to the Author

Reviewer #1: The overall study is good. However, the way the manuscript is written is a bit confusing at several places. Consider reframing some sections.

Reference 1 is not in English

Line 41 reducing up to to 65% the percentage of E.coli adhered

Line 222- The culture medium was changed when it changed color (how often was that? please explain why)

Line 281- A 200 ul aliquot

Reference where in the manuscript is the data for the actual numbers that were counted after the incubation mentioned below.

Line 233- The number of

234 adhered bacteria was determined by adding 200 μl of 0.5% sodium deoxycholate

235 (Sigma-Aldrich) to detach the bacteria from the cells. Detached bacteria were 10-

236 fold serially diluted, plated on MH agar, and counted after incubation at 37°C for

237 48 h. The adherence pattern was determined by Giemsa staining. Samples of

238 adhered and non-adhered bacteria were collected for gene expression assays.

The oregano extracts can not be consistent every time when prepared. What do the authors do the make sure the extracts are not too different each time prepared?

6. PLOS authors have the option to publish the peer review history of their article (what does this mean?). If published, this will include your full peer review and any attached files.

Reviewer #1: No

---

## [Author Response · Author response to Decision Letter 0]

11 Jan 2021

PLOS ONE

Attn: Dr. Yung-Fu Chang

Academic Editor 

Re: Response to reviewer comments for PONE-D-20-36066

Dear Dr. Chang:

Thank you for comments to PONE-D-20-36066, “Natural and synthetic antimicrobials reduce adherence of enteroaggregative and enterohemorrhagic Escherichia coli to epithelial cells”. We detail below an itemized numbered list stating (in italics) how we addressed each of the reviewers' concerns in a revised manuscript. 

R= MS was checked according to the journal instructions 

 2.) We note that you have included the phrase “data not shown” in your manuscript. Unfortunately, this does not meet our data sharing requirements. PLOS does not permit references to inaccessible data. We require that authors provide all relevant data within the paper, Supporting Information files, or in an acceptable, public repository. Please add a citation to support this phrase or upload the data that corresponds with these findings to a stable repository (such as Figshare or Dryad) and provide and URLs, DOIs, or accession numbers that may be used to access these data. Or, if the data are not a core part of the research being presented in your study, we ask that you remove the phrase that refers to these data.

R= The legend was deleted since this finding was not relevant to this research

 3.)PLOS ONE now requires that authors provide the original uncropped and unadjusted images underlying all blot or gel results reported in a submission’s figures or Supporting Information files. This policy and the journal’s other requirements for blot/gel reporting and figure preparation are described in detail at https://journals.plos.org/plosone/s/figures#loc-blot-and-gel-reporting-requirements and https://journals.plos.org/plosone/s/figures#loc-preparing-figures-from-image-files. When you submit your revised manuscript, please ensure that your figures adhere fully to these guidelines and provide the original underlying images for all blot or gel data reported in your submission. See the following link for instructions on providing the original image data: https://journals.plos.org/plosone/s/figures#loc-original-images-for-blots-and-gels.

R= Image files were not manipulated or adjusted in any way. Pictures were taken directly from microscope. 

 4.) Please upload a new copy of Figure 1 as the detail is not clear. Please follow the link for more information: https://blogs.plos.org/plos/2019/06/looking-good-tips-for-creating-your-plos-figures-graphics/" https://blogs.plos.org/plos/2019/06/looking-good-tips-for-creating-your-plos-figures-graphics/

R= An improved fig 1 was included. 

Reviewer #1 (Remarks to Author): 

Reviewer #1: The overall study is good. However, the way the manuscript is written is a bit confusing at several places. Consider reframing some sections.

Reference 1 is not in English

R= Done

Line 41 reducing up to to 65% the percentage of E.coli adhered

R= the words “the percentage” were deleted in text. 

Line 222- The culture medium was changed when it changed color (how often was that? please explain why)

R= Media was changed at 48-72 h. Media changed of color due to acidification of pH indicator as consequence of cellular growth. This information was included in the text. 

Line 281- A 200 ul aliquot

R= Done

Reference where in the manuscript is the data for the actual numbers that were counted after the incubation mentioned below.

Line 233- The number of

234 adhered bacteria was determined by adding 200 μl of 0.5% sodium deoxycholate

235 (Sigma-Aldrich) to detach the bacteria from the cells. Detached bacteria were 10-

236 fold serially diluted, plated on MH agar, and counted after incubation at 37°C for

237 48 h. The adherence pattern was determined by Giemsa staining. Samples of

238 adhered and non-adhered bacteria were collected for gene expression assays.

R= Data appear in Figure 1 which was improved.

The oregano extracts can not be consistent every time when prepared. What do the authors do the make sure the extracts are not too different each time prepared?

R= The same batch of oregano was always used. It was preserved in dry leaves, and under controlled conditions of light (darkness), humidity and temperature. When the fresh extract was prepared, the MBC (minimum bactericidal concentration) was determined; no variations on MBC was found. This information was included in the text.

Sincerely Yours:

Norma L. Heredia

---

## [Decision Letter · Decision Letter 1]

25 Mar 2021

PONE-D-20-36066R1

Natural and synthetic antimicrobials reduce adherence of enteroaggregative and enterohemorrhagic Escherichia coli to epithelial cells

PLOS ONE

Dear Dr. Heredia,

Thank you for submitting your manuscript to PLOS ONE. After careful consideration, we feel that it has merit but does not fully meet PLOS ONE’s publication criteria as it currently stands. Therefore, we invite you to submit a revised version of the manuscript that addresses the points raised during the review process.

Your manuscript has been returned to the original reviewer and a minor revision  is still suggested.

We look forward to receiving your revised manuscript.

Kind regards,

Yung-Fu Chang

Academic Editor

PLOS ONE

Journal Requirements:

Reviewers' comments:

Reviewer's Responses to Questions

**Comments to the Author**

1. If the authors have adequately addressed your comments raised in a previous round of review and you feel that this manuscript is now acceptable for publication, you may indicate that here to bypass the “Comments to the Author” section, enter your conflict of interest statement in the “Confidential to Editor” section, and submit your "Accept" recommendation.

Reviewer #1: All comments have been addressed

2. Is the manuscript technically sound, and do the data support the conclusions?

Reviewer #1: Yes

3. Has the statistical analysis been performed appropriately and rigorously? 

Reviewer #1: Yes

4. Have the authors made all data underlying the findings in their manuscript fully available?

Reviewer #1: Yes

5. Is the manuscript presented in an intelligible fashion and written in standard English?

Reviewer #1: Yes

6. Review Comments to the Author

Reviewer #1: The authors needs to check the italics on gene names. It isn't consistent throughout the manuscript.

7. PLOS authors have the option to publish the peer review history of their article (what does this mean?). If published, this will include your full peer review and any attached files.

Reviewer #1: No

---

## [Author Response · Author response to Decision Letter 1]

29 Mar 2021

PLOS ONE

Attn: Dr. Yung-Fu Chang

Academic Editor 

E-mail: em@editorialmanager.com

Re: Response to reviewer comments for PONE-D-20-36066R1

Dear Dr. Chang:

Thank you for comments to PONE-D-20-36066R1, “Natural and synthetic antimicrobials reduce adherence of enteroaggregative and enterohemorrhagic Escherichia coli to epithelial cells”. We detail below an itemized numbered list stating (in italics) how we addressed each of the reviewers' concerns in a revised manuscript. 

R= All references were checked. No cited paper has been retracted

Reviewer #1 (Remarks to Author): 

Comments to the Author

The authors need to check the italics on gene names. It isn't consistent throughout the manuscript.

R= In this manuscript we referred to Shiga toxin 1 protein and/or Shiga toxin 2 protein as Stx1 and Stx2 respectively (these are not genes). We slightly modified the text to clarify this. 

Sincerely Yours:

Norma L. Heredia

---

## [Decision Letter · Decision Letter 2]

20 Apr 2021

Natural and synthetic antimicrobials reduce adherence of enteroaggregative and enterohemorrhagic Escherichia coli to epithelial cells

PONE-D-20-36066R2

Dear Dr. Heredia,

We’re pleased to inform you that your manuscript has been judged scientifically suitable for publication and will be formally accepted for publication once it meets all outstanding technical requirements.

Kind regards,

Yung-Fu Chang

Academic Editor

PLOS ONE

Additional Editor Comments (optional):

Reviewers' comments:

Reviewer's Responses to Questions

**Comments to the Author**

1. If the authors have adequately addressed your comments raised in a previous round of review and you feel that this manuscript is now acceptable for publication, you may indicate that here to bypass the “Comments to the Author” section, enter your conflict of interest statement in the “Confidential to Editor” section, and submit your "Accept" recommendation.

Reviewer #1: All comments have been addressed

2. Is the manuscript technically sound, and do the data support the conclusions?

Reviewer #1: Yes

3. Has the statistical analysis been performed appropriately and rigorously? 

Reviewer #1: Yes

4. Have the authors made all data underlying the findings in their manuscript fully available?

Reviewer #1: Yes

5. Is the manuscript presented in an intelligible fashion and written in standard English?

Reviewer #1: Yes

6. Review Comments to the Author

Reviewer #1: (No Response)

7. PLOS authors have the option to publish the peer review history of their article (what does this mean?). If published, this will include your full peer review and any attached files.

Reviewer #1: No